# GRP94 Inhabits the Immortalized Porcine Hepatic Stellate Cells Apoptosis under Endoplasmic Reticulum Stress through Modulating the Expression of IGF-1 and Ubiquitin

**DOI:** 10.3390/ijms232214059

**Published:** 2022-11-14

**Authors:** Xiaohong Wang, Hairui Xin, Chuang Zhang, Xianhong Gu, Yue Hao

**Affiliations:** State Key Laboratory of Animal Nutrition, Institute of Animal Sciences, Chinese Academy of Agricultural Sciences, Beijing 100193, China

**Keywords:** endoplasmic reticulum stress, piglet liver, hepatic stellate cell, GRP94, IGF-1, ubiquitin, apoptosis

## Abstract

Endoplasmic reticulum stress (ERS) is closely related to the occurrence and progression of metabolic liver disease. The treatment targeting glucose-regulated protein 94 (GRP94) for liver disease has gotten much attention, but the specific effect of GRP94 on hepatocyte apoptosis is still unclear. So far, all the studies on GRP94 have been conducted in mice or rats, and little study has been reported on pigs, which share more similarities with humans. In this study, we used low-dose (LD) and high-dose (HD) tunicamycin (TM) to establish ERS models on piglet livers and immortalized porcine hepatic stellate cells (HSCs). On the piglet ERS model we found that ERS could significantly (*p* < 0.01) stimulate the secretion and synthesis of insulin-like growth factor (IGF-1), IGF-1 receptor (IGF-1R), and IGF-binding protein (IGFBP)-1 and IGFBP-3; however, with the increase in ERS degree, the effect of promoting secretion and synthesis significantly (*p* < 0.01) decreased. In addition, the ubiquitin protein and ubiquitination-related gene were significantly increased (*p* < 0.05) in the LD group compared with the vehicle group. The protein level of Active-caspase 3 was significantly increased (*p* < 0.01) in the HD group, however, the TUNEL staining showed there was no significant apoptosis in the piglet liver ERS model. To explore the biofunction of ER chaperone GRP94, we used shRNA to knock down the expression of GRP94 in porcine HSCs. Interestingly, on porcine HSCs, the knockdown of GRP94 significantly (*p* < 0.05) decreased the secretion of IGF-1, IGFBP-1 and IGFBP-3 under ERS, but had no significant effect on these under normal condition, and knockdown GRP94 had a significant (*p* < 0.01) effect on the UBE2E gene and ubiquitin protein from the analysis of two-way ANOVA. On porcine HSCs apoptosis, the knockdown of GRP94 increased the cell apoptosis in TUNEL staining, and the two-way ANOVA analysis shows that knockdown GRP94 had a significant (*p* < 0.01) effect on the protein levels of Bcl-2 and Caspase-3. For CCK-8 assay, ERS had a significant inhibitory(*p* < 0.05) effect on cell proliferation when treated with ERS for 24 h, and both knockdown GRP94 and ERS had a significant inhibitory(*p* < 0.05) effect on cell proliferation when treated with ERS for 36 h and 48 h. We concluded that GRP94 can protect the cell from ERS-induced apoptosis by promoting the IGF-1 system and ubiquitin. These results provide valuable information on the adaptive mechanisms of the liver under ERS, and could help identify vital functional genes to be applied as possible diagnostic biomarkers and treatments for diseases induced by ERS in the future.

## 1. Introduction

The endoplasmic reticulum (ER) is responsible for the synthesis and depletion of proteins, carbohydrates and fats, playing an important role in cell metabolism. When the body is faced with physical or chemical stimulations, which can lead to the accumulation of unfolded proteins in ER stress (ERS), ERS will be stimulated with the disturbance of calcium, and this may be the earliest stress response [1]. Under ERS, the cell will stimulate the unfolded protein response (UPR), and then the three UPR sensor proteins inositol requiring enzyme 1 (IRE1), activating transcription factor 6 (ATF6) and PKR-like eukaryotic initiation factor 2α kinase (PERK) will be activated to mediate the downstream pathways. The activation of UPR leads to the upregulation of pro-survival proteins involved in angiogenesis, folding capacity, redox protection, or the degradation of unfolded proteins; however, prolonged UPR can also lead to apoptosis [2,3]. As such, slight ERS will be handled by the body easily, but long-lasting and strong ERS will lead to cell apoptosis, which is the landmark of organ injury [4]. In the liver, many studies have proven that ERS is associated with the pathogenesis of many liver diseases [5]. ERS always occurs with liver steatosis [6], inflammation [7], and apoptosis [8], which lead to the deterioration of acute or chronic liver disease [9], such as non-alcoholic fatty liver disease (NAFLD), acute liver failure (ALF) [10], and cirrhosis [11]. In certain types of liver injury, hepatic apoptosis is considered as a leading pathological characteristic. Intervening in hepatic apoptosis is likely to stop the spread of disease and lower the morbidity of liver disease [12]. So, regulating the ERS process has becoming a significant approach in liver disease treatment. It has been reported that ERS-mediated apoptosis is of great importance in the treatment of liver fibrosis [13,14], and Torres et al. found that administrating tauroursodeoxycholic acid to inhabit ERS can protect mice from hepatoxicity following ALF-inducer administration [15]. However, currently, no therapy strategy is being used satisfyingly in clinical practice. Investigations on the mechanism of hepatic apoptosis are still ongoing [16]. At present, gene therapy for liver disease has attracted much attention [17]. Glucose regulated protein 94 (GRP94) has gotten much attention as a target for treatment [18,19]. GRP94, also known as HSP90B1 (Heat shock protein), is a molecular chaperone of the HSP90 family and is expressed in a large number via the activation of UPR. Though the upregulation of GRP94 is always seen as a hallmark response to ERS, it is still undetermined whether GRP94 protects cells [20] or inhibits cell death [21]. Therefore, what effect GRP94 has on apoptosis and how GRP94 regulates apoptosis under ERS are still unclear. It is of great significance to investigate the molecular mechanisms of GRP94 modulating the apoptosis in hepatic cells under ERS, which will improve the theoretical basis for liver disease treatment.

The insulin-like growth factor (IGF-1) is a kind of polypeptide hormone that can promote cell division and differentiation, and inhibit apoptosis to resist stress. IGF-1 is produced primarily by the liver, and its function is achieved by binding to the IGF-1 receptor (IGF-1R), but this process is regulated by IGF-binding proteins (IGFBP), with IGFBP-3 carrying a majority of circulating IGF-1 [22]. IGF-1 is closely related to ERS; ERS may be altered by IGF-1 by increasing C/EBP homologous protein (CHOP) expression and IGF-1R, and so on [23]. IGF-1 can also induce the expression of UPR promoter GRP78, and knockdown IGF-1R represses GRP78 expression [24]. Many experiments have also found that IGF-1 will be highly expressed under several stress conditions, including heat stress [25], oxidative stress [26] and ERS [27]. Besides that, there is an intersection between these different stress forms; Cui et al. found that in pigs, chronic heat stress will also lead to ERS, with significantly increased GRP94 and IGF-1 [28]. However, the effect of ERS on IGF-1 remains controversial, and many experiments have also shown that ERS will suppress the axis of IGF-1 during stress [29,30].As such, we need to investigate the role of IGF-1 under ERS, and whether the ER chaperone GRP94 regulates IGF-1.

When ERS happens, the misfolded and unfolded proteins will eventually be reverse-transported from the ER to the cytoplasm, and then be ubiquitinated and degraded. This procedure is called endoplasmic reticulum-associated degradation (ERAD). In ERAD, the ubiquitin–proteosome system (UPS) is activated to degrade the unfolded and misfolded proteins labeled with ubiquitin to restore body homeostasis. Ubiquitin, ubiquitin-activating enzymes (E1), ubiquitin-conjugating enzymes (E2), and ubiquitin–protein ligase (E3) are all components of the UPS, and they work together to mono- or poly-ubiquitinate and degrade proteins [31]. ERAD plays an essential role in restoring homeostasis; the enhanced ERAD can significantly decrease cell apoptosis [32], and similarly, the impairment or inhabitation of ERAD will lead to cell apoptosis [33,34]. Therefore, various therapeutic approaches to UPR, ERAD, and UPS have been used to correct pathological diseases involving protein misfolding. So, we want to investigate whether GRP94, which acts as an ER chaperone, is correlated with ubiquitination.

In this experiment, acute ERS model was established in piglets, using tunicamycin (TM), which is an internationally recognized ERS inducer. It can induce fat accumulation in the liver, thus inducing NAFLD or NASH [35,36]. The piglets share more similarities with humans than mice and rats in terms of liver morphology and function [37], so pigs (Sus scrofa) are emerging as an attractive biomedical model for studying liver diseases in humans, because of their similar metabolic features and functions. Moreover, we used SV40LT overexpression lentivirus to immortalize the porcine hepatic stellate cells (HSCs) as the cell model. HSCs will transform from the quiescent state to activation under stress [38,39], which plays an important role in liver disease development [40]. Additionally, Mannaerts et al. found that UPR is involved in the early phase of HSCs activation [41]. Many studies have reported that upon ERS, HSCs can induce cell apoptosis [13,42], and arrest the cell cycle [43]. In rats with liver fibrosis, ERS can even lead to HSCs apoptosis with fibrosis resolution [14], so HSCs are suitable for ERS study. Considering that the retroviral vector can only infect dividing cells and has a limited capacity, we used lentivirus to inhibit the expression of GRP94 in immortalized porcine HSCs, and investigated the effect of low GRP94 expression on cell injury, ubiquitination and the IGF-1 system under ERS.

## 2. Results

### 2.1. Establishment of an ERS Model in Piglets

To evaluate ERS development and detect the expression levels of hepatic ERS-associated genes and proteins in piglets, we performed RT-PCR and western blot analyses. Protein levels of GRP78, GRP94 and p-IRE1 were evaluated in liver samples, as shown in Figure 1A–D. The GRP78, GRP94 and IRE1α protein levels were significantly increased (*p* < 0.01) in both the LD and HD groups, and the p-eIF2α protein expression level was significantly increased (*p* < 0.01) in the HD group compared with that in the vehicle group. The activation of ERS-linked genes, including GRP78, GRP94, ATF4, and ATF6, is shown in Figure 1F–I. We found that, compared with levels in the control group, the gene expression level of GRP78 was significantly upregulated (*p* < 0.01) in both the HD and LD groups, the gene expression level of GRP94 was significantly upregulated in both the HD (*p* < 0.01) and LD (*p* < 0.05) groups, the gene expression level of ATF4 was significantly increased (*p* < 0.01) in the HD group, and the gene expression level of ATF6 was not significantly different. Our data indicate that low and high TM doses stimulate the development of ERS and UPR in piglets.

### 2.2. ERS-Dependent Changes in IGF-1 Efficiency of Piglet Livers and Plasmas

To estimate the effects of ERS on IGF-1 efficiency, we performed ELISA analysis to detect the levels of IGF-1, IGF-1R, IGFBP-1, and IGFBP-3 in piglet livers and plasma. The hepatic levels of IGF-1, IGF-1R, IGFBP-1, and IGFBP-3 are shown in Figure 2A–D. The hepatic expressions of IGF-1, IGF-1R and IGFBP-1 were significantly increased (*p* < 0.01) in both LD and HD groups compared with those in the vehicle group, and those expressions were significantly higher (*p* < 0.01) in the LD group compared with those in the HD group. The hepatic level of IGFBP-3 was significantly higher (*p* < 0.01) in the LD group compared with that in the vehicle and HD groups. The plasma levels of IGF-1, IGF-1R, IGFBP-1, and IGFBP-3 are shown in Figure 2E–H. The plasma expressions of IGF-1 (*p* < 0.01), IGF-1R (*p* < 0.01), IGFBP-1 (*p* < 0.05) and IGFBP-3 (*p* < 0.05) were significantly increased in the LD group compared with those in the HD group, and all those expressions were significantly increased in both the LD and HD groups (*p* < 0.01). Our results show that ERS stimulated the synthesis and secretion of IGF-1 and related proteins affecting IGF-1 biological activity.

### 2.3. ERS-Dependent Changes in Ubiquitination of Piglet Livers

Four ubiquitin-related genes were detected in piglet liver, including two ubiquitin-activating enzymes (UBA2, UBA3) and two ubiquitin-conjugating enzymes (UBE2E, UBE2I). The results show that only UBE2E gene expression was upregulated in the liver of HD piglets (*p* < 0.01), while UBA2 (*p* < 0.05), UBA3 (*p* < 0.05) and UBE2E (*p* < 0.01) in the liver of LD were significantly increased. In addition, the levels of ubiquitin protein in the liver were detected by Western blot. The results show that the expression of ubiquitin in liver was higher in the LD group compared with thoso in the vehicle group and HD group. To sum up, compared with the HD and vehicle groups, the LD group showed more obvious ubiquitination in the liver (Figure 3).

### 2.4. ERS-Dependent Changes in Hepatocellular Apoptosis of Piglet Livers

A TUNEL assay was used to detect the apoptosis of hepatic cells under ERS; the results are shown in Figure 4. The TUNEL staining results of hepatic cells in the vehicle, LD and HD groups are shown in A–C. The results of apoptosis rate (D) demonstrate there is no significant difference between the LD group and HD group. In addition, RT-qPCR and Western blot were used to detect the expressions of apoptosis-related genes (Bcl2, Bax, Fas) and protein (Active-Caspase 3). The results show that with the increase of TM concentration, it could promote the protein expression of Active-Caspase 3, but no significant apoptosis was found in tunel assay.

### 2.5. Immortalized the Porcine HSCs

The separated porcine HSCs are shown in Figure 5A–D; A and B show the newly isolated HSCs observed under a 100× objective lens (A) and under a 200× objective lens (B). When observed under an inverted microscope, the fresh porcine hepatic stellate cells were distributed as individuals with a round shape or an elliptical shape, a clear cell contour, an intact cell membrane, and lipid droplets contained in the cytoplasm. It can be seen that the separated cells of the present invention were normal porcine HSCs. C and D show the primary cultured 3–5 d HSCs observed under a 100× objective lens (C) and under a 200× objective lens (D). The cells grew adherently.

The immunofluorescence map of the control group is shown in E–F, where E is the immunofluorescence map of 100×–DAPI, and F is the immunofluorescence map of 100×–fluorescence. The immunofluorescence map of the test group is shown in G–H, where G is the immunofluorescence map of 100×–DAPI, and H is the immunofluorescence map of 100×–fluorescence. As shown in Figure 5E–H, immunofluorescence cytochemical staining was performed with an HSCs marker protein Desmin to identify the obtained cells as HSCs. Desmin is a protein unique to smooth muscle cells, also known as fibronectin. HSCs have the characteristics of myofibroblasts, and thus Desmin is stably present in primary and passaged stellate cells, that is, both the stationary and activated HSCs synthesized Desmin. Immunofluorescence showed that the cells presented as positive under Desmin staining.

The successfully transfected porcine HSCs of the first generation (Pl) were obtained after the screening, and the results are shown in I–J, where I is the map observed under a 100× objective lens, and J is the map observed under a 200× objective lens. It can be seen that the cells transfected with the SV40LT-overexpressing lentiviruses adhered to the wall, grew extended pseudopods, had clear cell contours, and had complete cell membranes. They had typical morphological features, which were similar to those of the primary porcine hepatic stellate cells.

The successfully transfected porcine hepatic stellate cells of the first generation (Pl) were passaged until the tenth generation (P10), and the results are shown in K and L, where K is the map observed under a 100× objective lens, and L is the map observed under a 200× objective lens. As can be seen, after 10 passages, the cells showed no significant differences in morphology and characteristics from the cells of the first generation.

### 2.6. Inhibition of GRP94 by GRP94-shRNA Lentivirus Transfection

Fluorescent protein-labeled plasmid vectors pHS-AVC-NC (negative control (NC)), pHS-AVC-1 (GRP94-shRNA1), pHS-AVC-2 (GRP94-shRNA2) and pHS-AVC-3 (GRP94-shRNA3) were synthesized and packaged into lentivirus by Shanghai GeneChem Co., Ltd. (Shanghai, China). The titer of the virus was as follows: shRNA-1—3.43 × 10^8^ TU/mL; shRNA-2—2.53 × 10^8^ TU/mL; shRNA-3—1.73 × 10^8^ TU/mL; NC—6.65 × 10^8^ TU/mL (as shown in Figure 6A). Porcine HSCs were transfected with GRP94-shRNA lentivirus at 37 °C and 5% CO_2_, respectively, and the brightness of red fluorescence was observed after 72 h. According to the results of PCR (Figure 6B) and Western blot (Figure 6C) analyses, the lentivirus shRNA-2 had a significant (*p* < 0.05) and stronger inhibitory effect on GRP94. Based on the results above, lentivirus shRNA-2 was selected for the follow-up experiment.

### 2.7. The Effect of GRP94 Knockdown and ERS on ERS Mark Genes and Proteins in HSCs

ERS molecule expressions were detected by RT-PCR or Western blot, as shown in Figure 7. The results show that the expressions of the GRP78 gene and protein, the GRP94 gene and protein, the ATF6 gene, the ATF4 gene and the IRE-1 gene were significantly (*p* < 0.05) increased under ERS, indicating TM applied to HSCs led to ERS. The GRP94 gene and protein were significantly (*p* < 0.05) decreased under lentivirus treatment compared with those in the vehicle group, especially under ERS, confirming the validity of lentivirus. Under ERS, the GRP78 gene was increased significantly (*p* < 0.05) in the GRP94 knockdown group compared with that in the vehicle group, and for the ATF4 gene, ATF6 gene, and IRE1 gene, GRP94 knockdown did not cause a significant effect. For the p-IRE1 protein, under ERS, GRP94 knockdown significantly (*p* < 0.05) decreased its expression compared with that in the NC group and the GRP94 knockdown group. Our results have been analyzed by two-way ANOVA, which not only verified the use of the ERS model on HSCs, but also found that knockdown GRP94 can affect IRE1 phosphorylation.

### 2.8. The Effect of GRP94 Knockdown and ERS on IGF-1 in HSCs

The cell supernatant was collected to detect IGF-1, IGF-1R, IGFBP-1, and IGFBP-3 expression using relevant kits, and the results are shown in Figure 8A–D. Under ERS, the expressions of all IGF-1, IGFBP-1 and IGFBP-3 were significantly increased (*p* < 0.05). Compared with vehicle group, the GRP94 knockdown group showed significantly (*p* < 0.05) lower expressions of IGF-1, IGFBP-1, and IGFBP-3 under ERS. The expressions of IGF-1R did not change significantly (*p* < 0.05) with ERS or shRNA. As such, in HSCs, ERS stimulated the secretion of IGF-1, IGFBP-1 and IGFBP-3, and knockdown GRP94 inhibited the secretion of those under ERS.

### 2.9. The Effect of GRP94 Knockdown and ERS on Ubiquitination in HSCs

The cells were collected to detect mRNA expressions of UBA2, UBA3, UBE2E, and UBE2I using RT-PCR, to detect the protein expressions of ubiquitin using Western blot, the related results are shown in Figure 9A–F. ERS significantly stimulated UBA2 gene expression (*p* < 0.05) in the vehicle and NC groups, with no difference in the GRP94 knockdown group. From the point of view of two-way ANOVA, ERS and GRP94 knockdown significantly (*p* < 0.01) increased the gene expression of UBE2E, and ERS significantly stimulated the gene expression of UBE2I (*p* < 0.05). Under ERS, the protein level of ubiquitin in the GRP94 knockdown group was significantly (*p* < 0.05) lower than that in the NC group. These results illustrate that ERS stimulated ubiquitin-related gene expressions and GRP94 knockdown increased the expression of ubiquitin protein.

### 2.10. The Effect of GRP94 Knockdown and ERS on Apoptosis in HSCs

The cells were collected to detect cell apoptosis by fluorescence staining and Western blot, and the results are shown in Figure 10. The images of cell fluorescence staining are shown in A–C, demonstrating that under TM the cell apoptosis rate increased significantly with GRP94 knockdown, suggesting that GRP94 plays a protective role in HSCs under ERS. After transfection with lentivirus, cells were treated with TM for 24 h, 36 h and 48 h for the CCK-8 assay to detect cell viability; the results are shown in D–F. In 24 h, ERS decreased the cell viability significantly (*p* < 0.05) in all groups, especially in the GRP94 knockdown group. In 48 h, ERS decreased the cell viability more noticeably; interestingly, regardless of whether under normal condition or ERS, the cell viability was significantly (*p* < 0.05) inhibited in the GRP94 knockdown group. The western blot results are shown in G–I. From the point of view of two-way ANOVA, GRP94 knockdown significantly (*p* < 0.01) affected the protein levels of Bcl-2 and Caspase 3 independently of ERS, further indicating the anti-apoptosis effect of GRP94. The qPCR results show that ERS significantly (*p* < 0.01) decreased the Caspase 3 and Bcl-2 mRNA expressions, and in terms of GRP94 knockdown, it seems that there was no significant change trend.

## 3. Discussion

GRP94 is one of the ER chaperones, and the liver, as the biggest metabolic organ, is rich in ER, and is the major location of synthesis of IGF-1 and IGFBPs [44]. Besides this, pigs share more similarities with human in terms of metabolism, and ERS has a close connection with the occurrence and development of various liver diseases, such as NAFLD and cirrhosis [45,46]. Therefore, to explore the effects of ERS and GRP94 on the IGF system, the ERAD process and apoptosis, we selected piglet livers as the animal model. The immortalized porcine HSCs that were not significantly different from the primary porcine HSCs were used as the cellular model for the study.

ERS is triggered by various factors to disturb the normal function of ER, due to the accumulation of unfolded and misfolded proteins resulting in the breaking of cell homeostasis. UPR is activated under ERS as an intracellular response. It can control protein quality as the first line of defense against the accumulation of unfolded and misfolded proteins in the ER lumen [4]. As such, treatments targeting UPR have gained much attention [47]. In UPR, GRP94 is highly expressed to participate in protein folding, interacting with other components of the ER protein folding machinery. Recently, the relation between GRP94 and cell apoptosis has attracted much attention, but the specific effect of GRP94 on apoptosis, and by which pathway to effect, are still unclear. Besides this, GRP94 is more selective than many ER chaperones, and the basis for this selectivity remains obscure [48].

TM is the classic inducer of ERS, which can inhibit cell cycle progression in vivo and in vitro by blocking N-linked glycosylation [42], and is already widely used in mice and rats, but there is little research reported on piglets, which have a closer relationship with humans in terms of the metabolism [37]. In this study, we used TM to establish an ERS model on piglet livers and HSCs, both of which were verified by the significantly increased levels of ERS marker gene and protein expressions. Both GRP78 and GRP94, as ERS-induced molecules, are ER chaperones, participating in protein folding, interacting with other components of the ER protein folding machinery, and assisting in the targeting of misfolded proteins [49]. GRP78 is more highly conserved than GRP94, and GRP78, associating the UPR sensor proteins IRE1, PERK and ATF6, acts as a gatekeeper of UPR. In this study, knockdown GRP94 under ERS significantly increased the GRP78 gene’s expression, indicating the compensatory synthesis of GRP78 under GRP94 deletion, which can explain why only reducing the expression of GRP94 does not affect the homeostasis and ERS of hepatocytes [50], and GRP94-deficient cells can grow normally [51], but the co-downregulation of GRP78 and GRP94 will induce cell apoptosis and inhibit cell migration [52]. The UPR sensor protein PERK is activated under ERS, and can phosphorylate the alpha subunit of eIF2 (p-eIF2α) to inhibit global translation initiation and protein synthesis. ATF6 is also one of the UPR arms; ERS or the dissociation of GRP78 lead to its cleavage, which activates the UPR target genes, including ER chaperones. The IRE1α-X-box binding protein 1 (XBP1) axis is the most conserved UPR branch, which rapidly senses ERS and can transcriptionally regulate UPR target genes to restore protein folding capability and ER-associated protein degradation [53]. In this study, GRP94 knockdown does not have a significant effect on ATF4 and ATF6, verifying that GRP94 is more selective than GRP78, but GRP94 knockdown inhibited p-IRE1 protein expressions under ERS. Studies have shown that GRP94 deletion leads to a decrease in XBP1 protein cleavage [54], and XBP1 is present downstream of IRE1. Combined with this reference and our experiment, we conclude that GRP94 can affect the phosphorylated IRE1 pathway under ERS.

UPR can enhance the bioavailability and content of IGF-1, which has anti-apoptosis and proliferation-promotion effects on cells. IGF-1 affects cells by binding to IGF-1R to trigger a conformation change and activate downstream signals. IGFBPs can extend IGF-1’s half-life, and are the principal regulator of IGF-1 bioactivity. IGFBP-1 is predominantly expressed in the liver, and can bind to IGF-1 to prevent it from binding to the IGF receptor and then inhibit IGF-1-stimulated cell growth and the differentiative function in an IGF-dependent way [55,56]. In the piglet ERS model, we found that IGF-1, IGF-1R, IGFBP-1 and IGFBP-3 share the same change trend in liver and plasma, the expressions of these molecules in the LD group were significantly higher than those in the HD group, and the expressions in the HD group were significantly higher than those in the vehicle group. Therefore, we suppose that under slight ERS, IGF-1 is increased to protect cells from survival, but IGF-1 will tend to decrease with increased ERS intensity due to the destruction of ER under long-lasting and strong ERS, which can explain the other findings of strong ERS inhibiting the synthesis of IGF-1 to induce growth retardation [29]. In normal condition, IGF-1R maintains low levels in the liver, but its expression will increase significantly when under pathological conditions (such as hepatitis) [57]; our results for IGF-1R verify this conclusion. Various ERS-inducers can increase the IGFBP-1 mRNA and secreted protein levels up to 20-fold [58]. IGFBP-1 displays tissue-specific expressions, and IGFBP-1 gene knockout studies in mice have suggested that in injured livers, IGFBP-1 may function as an IGF-independent pro-mitogenic and protective protein [59]. Therefore, in our study, the increased IGFBP-1 production during hepatic ERS may serve as a signal to regulate cell growth and trigger a systemic adaptive response. In the blood, 90% of IGF-1 is associated with IGFBP-3, which can attenuate the interaction of IGF-1 and IGF-1R [60]. Cui found that heat stress could lead to liver ERS [28], and Xin also found that under heat stress, the IGFBP-3 in pig plasma increased at first and then decreased [61]. On the one hand, the increased IGFBP-3 takes part in the regulation of IGF-1 bioavailability; on the other hand, it could be seen as a defense mechanism against the insulin-like hypoglycemia effect of IGF-1 when a pathological state is present [61]. IGF-1 is not only located downstream of UPR, but Argon has recently come to the conclusion that IGF-1 is regulated by GRP94 [62]. In our study, under ERS, knockdown GRP94 significantly decreased the secretion of IGF-1 and IGFBPs. The biological importance of the GRP94–IGF-1 interaction is highlighted by the discovery in the muscle of mice that knockdown GRP94 leads to decreased circulating IGF-1 and muscle weight [63]. Notably, in the absence of IGFBPs, there are much lower levels of serum IGF-1 [62], so in our study, under GRP94 knockdown, the decreased IGFBPs, which are responsible for the stability of IGF-1, may adapt to the decreased IGF-1. There is, however, little evidence that there is a relation between IGFBPs and GRP94.

The ubiquitin–proteasome system, which is activated by UPR in ERAD, is involved in the major protein degradation system [31,64]. Ubiquitin, a polypeptide chain consisting of 76 amino acids, will label protein misfolding to degrade it. Ubiquitin binds to target proteins through the synergy of E1 ubiquitin-activating enzymes, E2 ubiquitin-conjugating enzymes and E3 ubiquitin ligase enzymes. E1 can covalently bind and activate ubiquitin, UBA2 and UBA3 are both E1 ubiquitin-activating enzymes, E2 ubiquitin-conjugating enzymes are the center of three enzymes that bind ubiquitin to substrates during ubiquitination, and UBE2E and UBE2I are both E2 ubiquitin-conjugating enzymes, while E3 mediates the transfer of ubiquitin from the E2-binding enzyme to a specific ubiquitin [31]. In the piglet liver model, the changing trends of ubiquitin-activating enzymes, ubiquitin-conjugating enzymes and ubiquitin protein showed that there is a more significant process in the LD group than in the HD group. As such, we suppose that the ubiquitination process will also be reduced as the ERS intensity increases. In the HSCs model, GRP94 knockdown significantly decreased the ubiquitin protein level, demonstrating that GRP94 is closely related to the ubiquitination process. This may be because GRP94 as an ER chaperone is responsible for transporting misfolded proteins to the cytoplasm to degrade. Furthermore, the GRP94 gene co-expression network, identified by Wang et al., suggests the close relation of GRP94 and ubiquitin [65], and studies have found that GRP94 can mediate ERAD by delivering the γ-aminobutyric acid type A receptor and protein OS-9 [66,67]. Among the three UPR sensor proteins, IRE1α is the most commonly studied, and the ubiquitin ligases for ATF6 and PERK have not been found, but IRE1α is reported to be regulated by multiple enzymes related to ubiquitination [31]. When IRE1α-XBP1 is triggered to modulate protein synthesis under ERS, it can further activate the ERAD process to alleviate the ER burden [68,69]. However, once the stress level exceeds the handling threshold of the cell, the IRE1α signaling axis decays gradually, and promotes the activation of the apoptosis-related pathway, causing cell apoptosis [70,71,72]. Combined with the previous results on the relation between IRE1α and GRP94, we conclude that knockdown GRP94 under ERS will block the IRE1α signal pathway, and then affect the ERAD process, which is characterized by a decreased degree of ubiquitination. As a complicated process with many proteins and enzymes involved, the specific ubiquitination mechanism under ERS still requires further exploration.

The most important function of GRP94 as an ER chaperone is directing folding or assembling the unfolded protein, but the significance of GRP94 in apoptosis has been debated. Its ability to protect cells seems to depend on the type of stress. Ma found that downregulating GRP94 expression with miR-150 protected against hypoxia-induced apoptosis [73]. In vascular endothelial cells, Wei found that inhibiting the expression of GRP94 inhibits autophagy and apoptosis induced by oxidized low-density lipoproteins [74]. Conversely, scholars have also found that the anti-apoptosis effect of GRP94, induced by silencing GRP94 by siRNA, increased pancreatic cancer cells apoptosis in vitro [75]. In keratinocytes, the upregulation of GRP75, GRP78 and GRP94 may be able to prevent protein unfolding, to counter ERS and to inhibit cell apoptosis [76]. Our results of Tunnel staining and the factors related to apoptosis in piglet livers show that an intraperitoneal injection of TM dose-dependently promoted the Active-capase 3 protein expression, but did not cause significant cell apoptosis. Zhu used apoptin to prove that enough ERS can damage the ER structure, but not affect ERS [77]. Bian et al. also found that the cell apoptosis induced by TM is dose-dependent [78]. Therefore, 0.1 mg/kg and 0.3 mg/kg TM did not cause severe ERS, which may be why no significant apoptosis was observed on ERS piglet livers, and another possible reason may be that the body has a more comprehensive regulatory function than the cell, so it has a stronger defense effect against ERS. In HSCs, the Tunnel staining results (A–C) revealed that under ERS, GRP94 knockdown increased cell apoptosis clearly. The function of GRP94 under ERS is self-evident. The cell relieves ERS mainly in two ways; on the one hand, UPR is activated, three sensor proteins are initiated to reduce protein synthesis and accumulation [4], and the most conserved sensor protein IRE1α can slow down the synthesis of new polypeptide chains to reduces ER pressure. The knockdown of GRP94 affects the activation of the IRE1α pathway, which has been proven in our experiments, so the knockdown of GRP94 cannot effectively alleviate ERS. On the other hand, the misfolded proteins presenting under ERS will be degraded by the ERAD process, but the loss of GRP94 impacts the ubiquitination system, which has been proven in our experiments, resulting in the accumulation of misfolded proteins. Interestingly, the CCK-8 test showed that GRP94 knockdown decreased cell viability; the longer the time, the more obvious the effect, not only under ERS. Furthermore, with the extension of time, the knockdown of GRP94 even significantly affected cell viability under normal condition. Our results for normal condition are consistent with those of previous studies [79], suggesting that GRP94 also has a part to play in cell proliferation. In cell apoptosis, the qPCR results of Bcl-2 and Caspase-3 contain controversies. In terms of gene expressions, ERS significantly lowered the expression of Bcl-2, indicating the pro-apoptosis effect of ERS, but the change in Caspase-3 gene expression shows that ERS has a tendency to inhibit cell apoptosis, demonstrating there may be a negative feedback effect of HSCs on preventing the constant cell apoptosis caused by ERS. As for the shRNA effect, no obvious trend was observed, which might suggest that, compared with GRP94 expression, ERS may be the more important factor in cell fate. However, the protein is the basic unit of biological function execution. Therefore, we should focus more on the protein changes of apoptosis-related factors. The protein expressions of anti-apoptosis factor Bcl-2 show a downward trend, while the protein expressions of pro-apoptosis factor Caspase-3 show an upward trend, and these results are consistent with the previous report of GRP94 overexpression, which resulted in the expression of anti-apoptotic proteins and blocked apoptosis [80], supporting the anti-apoptotic effect of GRP94. Notably the two-way ANOVA analysis showed that these protein changes of apoptosis-related factors caused by GRP94 knockdown do not interact significantly with ERS, which can account for the targeted deletion of GRP94 in the livers of mice, resulting in liver injury under normal condition [18]. Furthermore, GRP94 will be expressed highly under ERS; expression change proves the molecule’s function, so we used shRNA to knock down its expression to explore its function. GRP94 also showed anti-apoptosis effects similar to those under normal condition, which indicates the susceptibility of the cell to the expression of GRP94 under ERS. Combined with the above results, we can conclude that GRP94 plays an essential role in cell fate, both under ERS and normal condition. Overall, our results elaborate on the mechanism of GRP94 under ERS, as shown in Figure 11. A large number of unfolded proteins in the ER lumen under ERS promoted the dissociation of GRP94 from the compound, thus binding one of the UPR sensor proteins to IRE1α to promote its phosphorylation and activate the downstream pathway. Ultimately, this protects cells from stress by enhancing the IGF-1 system and the ERAD process.

## 4. Materials and Methods

### 4.1. Animals and Establishment of ERS Model

Twenty-one 35-day-old Duroc three-way crossbred barrows weighing 9.34 ± 0.3 kg were selected from 7 litters and raised in an environmentally controlled cabin maintained at 28 °C and 55% humidity. After 1 week of environmental adaptation, to eliminate differences in genetic background, three full siblings per litter were allocated to three groups as follows: (1) vehicle-treated group (5% dimethyl sulfoxide (DMSO), intraperitoneal injection); (2) TM low dosage (LD)-treated group (intraperitoneal injection, 0.1 mg/kg body weight (bw)); (3) TM high dosage (HD)-treated group (intraperitoneal injection, 0.3 mg/kg bw). Each piglet was raised in a single cage, and 7 piglets from the same treatment group were raised in one environmental control cabin. TM was dissolved in 5% DMSO and diluted with saline to obtain the desired concentration. All treated piglets were free to feed and drink.

Jugular venous blood was collected from each pig via venipuncture using 10 mL BD Vacutainers (BD Biosciences, San Jose, CA, USA) containing sodium heparin. The collected blood was centrifuged at 3000× *g* for 15 min at 4 °C. The supernatant plasma was collected and then stored in 1 mL Eppendorf tubes at −20 °C until required. Prior to the planned end of the experiment, euthanasia criteria were established using a head-only electric stun tong apparatus (Xingye Butchery Machinery Co., Ltd., Changde, China) 48 h after the intraperitoneal injection of TM or vehicle. The left lateral lobe of each liver was dissected, snap-frozen in liquid nitrogen, and stored at −80 °C for mRNA and protein analysis.

### 4.2. Preparation of Porcine Immortalized HSCs

#### 4.2.1. Separation and Culture of Porcine Hepatic Stellate Cells

A pig was bled to death from the neck artery, and a part of the liver tissue was taken and placed into PBS containing P/S. We removed the envelope and other similar parts, and then the hepatic tissue was chopped into 12 pieces and washed in PBS at least 3 times. We digested the tissue with 0.1% collagenase type IV at a volume at least 3 times greater than the tissue volume in a 37 °C water bath for 1 h, then passed the digested tissue through a 200-mesh screen to obtain a whole liver cell suspension. We then transferred the suspension into a centrifuge tube and centrifuged at 400 rpm for 10 min to obtain the precipitated cells as parenchymal hepatic cells, transferred the supernatant into another centrifuge tube and centrifuged at 1500 rpm for 10 min, and the supernatant was discarded after completion of the centrifugation to obtain the precipitate as the porcine HSCs. We then resuspended the precipitate in a medium, transferred it into a T-25 flask, and cultured it in a 37 °C incubator for 60 min, and then the non-adherent cells were transferred into another new flask for culturing, where the cells cultured in this flask were HSCs.

#### 4.2.2. Immunofluorescence Identification

A 24-well plate was used to hold three glass slides, and each well was filled with 1 mL of a medium and 0.02 million cells/well. The cells were placed in an incubator for 2 h. After the cells had grown fully on the slides or covered glass, the medium was pipetted, and the cells were washed with PBS 1 time and fixed at 4 °C for 30 min by the addition of 4% PFA. The cells were washed with PBS for 3 × 5 min/time. It was also possible not to pipette the PBS the last time and let the mixture cool overnight at 4 °C. Moisture was removed from the glass slide and then the glass slide was placed onto a petri dish support. Then, dropped 50 μL of the membrane-rupturing confining solution (0.5% Trition X-100 mixed with PBS at 1:1, plus 10% serum) onto a waterproof membrane, and the side of the glass slide with cells was covered for 2 h. Preparation of primary antibody: the antibody was mixed with PBS at 1:100 for the dilution of the membrane-rupturing confining solution, then 50 μL of the primary antibody was dropped onto the waterproof membrane, and the slide was covered and kept at 4 °C. The secondary antibody was incubated at room temperature with protection from light (secondary antibody: PBS = 1:500) for 2 h, washed with PBS for 3 × 5 min/time, stained with DAPI (DAPI: PBS = 1:1000) for 5 min, and washed with PBS for 3 × 5 min/time. One drop of Fluoromount-G was dropped onto each glass slide, and the side of the glass slide with cells was covered. Experiments after digestion were divided into 3 groups, which were respectively a control group, an experimental control group wherein the primary antibody was replaced with PBS or BSA, and an experimental group wherein the primary antibody was Desmin.

#### 4.2.3. Lentivirus Transfection

The porcine HSCs were inoculated into a 6-well plate, with the number of cells per well being about 1 × 10^5^. On the next day, the medium was replaced after the cells had adhered to the walls of the wells, and we added 1 mL of a complete medium plus 20 μL SV40LT-overexpressing lentivirus. After 12 h, the cell state was observed and the medium was replaced with a fresh medium, and when the cells grew fully over the bottom of the plate, they were passed into a T25 culture flask. After transfection, the cells were expanded for 3–4 generations and then subjected to screening.

#### 4.2.4. Screening of Porcine HSCs

Determination of a killing curve: The untransfected porcine HSCs were plated into a 24-well plate at 0.05 million per well and incubated overnight. On the next day, the old medium was removed from the 24-well plate, and fresh media each containing different concentrations of puromycin (1 μg/mL, 2 μg/mL, 3 μg/mL, 4 μg/mL, 5 μg/mL, 6 μg/mL and 7 μg/mL) were added into the 24-well plate already plated with the cells. The medium was replaced with a fresh screening medium every 2 days. The survival rate of the cells was observed every day, and the minimum concentration of puromycin used was the lowest screening concentration that killed all cells within 1–4 d from the start of puromycin screening. The puromycin screening of transfected cells: On the first day, the transfect porcine HSCs were plated into a 24-well plate at 0.05 million per well and incubated overnight; on the next day, we removed the old medium from the 24-well plate, then added a complete medium containing puromycin (3 μg/mL) for incubating, replaced the medium with a fresh screening medium every 2 days, and observed the survival rate of the cells every day. Cells that survived at the same time point (3 d) were successfully transfected porcine hepatic stellate cells, and used to expand the screened cells.

#### 4.2.5. The Morphology of Immortalized Porcine HSCs

We put the Fisherbrand Coverglass for Growth in a 6-well plate, inoculated 1 × 10^5^ immortalized porcine HSCs in each well, and cultured this with a 37 °C incubator. After 24 h, we fixed the immortalized porcine HSCs with 2.5% pentanediol solution, and observed the cell morphology under light microphage.

### 4.3. Plasmid Construction and Transfection

Fluorescent protein-labeled plasmid vectors targeting GRP94 were synthesized and packaged into lentivirus by Shanghai GeneChem Co., Ltd. shRNA sequences are shown in Table 1. For the transfection experiments, we prepared the cell suspension, and used blood cell count board to count. We used complete medium (10%FBS + ICELL stellate cell basic medium + 1% double antibody + 1% growth factor) to adjust the cell concentration to 50,000 cells/mL. Each well was inoculated with 100 μL of cell suspension in a 96-well plate, with 6 holes in each group. These were cultured overnight in a 5% CO_2_ incubator at 37 °C. When the degree of cell convergence reached 50%, lentivirus infection began. The lentivirus was diluted with ICELL stellate cell basic medium at 1:100, and was cultured at 37 °C in a 5% CO_2_ incubator. After overnight culturing, we changed the medium to 100 μL of complete culture medium. After 72 h of culturing, we changed the medium to the corresponding blank medium or a medium containing TM (5 μg/mL). After treating with TM for 24 h, the culture medium was removed and the cells were collected.

### 4.4. TUNEL Assay

The 1 cm^3^ samples of the piglets’ left liver tissue were fixed with paraformaldehyde solution for at least 24 h, then embedded in paraffin wax after dehydration. The liver tissue was sliced to a thickness of 12 μm. The apoptosis was detected according to the steps of the TUNEL assay kit (Beyotime Institute of Biotechnology, Shanghai, China), the images were collected under a 400× microscope, and the number and rate of positive cells were recorded.

On the porcine HSCs, we used the TUNEL assay kit (Promega, Madison, WI, USA) to detect apoptosis. Briefly, the cell climbing slices were placed in a six-well plate, cultured for 24 h, then transfected with lentivirus, and 72 h later, the ERS group was treated with TM (5 μg/mL) for 24 h. The glass slides were fixed in 4% paraformaldehyde at room temperature for 25 min, incubated with 0.1% Triton X-100 for 10 min, and washed three times with PBS. Proteinase K solution (100 μL at a concentration of 20 μg/mL) was added to each slide, which was then incubated at room temperature for 20 min. Next, the nuclei were stained with DAPI for 5 min in the dark, then the samples were washed with PBST. Fluorescence images were obtained using a fluorescence inverted microscope.

### 4.5. Immunofluorescence Identification

Total protein was isolated from the frozen liver samples and immortalized porcine HSCs after being treated with cell lysate. Protein concentrations were measured using a bicinchoninic acid (BCA) assay (Beyotime Institute of Biotechnology, Shanghai, China). Hepatic protein samples were analyzed by Western blotting, as described previously [81]. Protein samples were separated by electrophoresis on 10% SDS-PAGE gel and transferred to a polyvinylidene difluoride membrane (Millipore, Billerica, MA, USA). The blotted membrane was incubated in 1×TBST blocking solution for 2 h at room temperature with 5% skimmed milk, washed, and then incubated with specific primary antibodies according to the experimental design overnight at 4 °C. The following antibodies were used: anti-glucose-regulated protein 78 (GRP78) (1:1000 dilution; SC-13968; Santa Cruz Biotechnology, Dallas, TX, USA), anti-glucose-regulated protein 94 (GRP94) (GRP94, 1:20,000 dilution; ab2791; Abcam, Cambridge, UK), anti-IRE1(1:4000 dilution; NB100-2324; Novus Biological, Littleton, CO, USA), anti-phosphorylated-IRE1 (p-IRE1) (1:4000 dilution; NB100-2323; Novus Biological, Littleton, CO, USA), anti-eukaryotic translation initiation factor 2α (eIF2α) (1:1000 dilution; ab235147; Abcam, Cambridge, UK), anti-phosphorylated-eukaryotic translation initiation factor 2α (p-eIF2α) (1:1000 dilution; 3398; Cell Signaling Technology, Danvers, MA, USA), anti-active-Caspase 3 (1:1000 dilution; 19677-1-AP; proteintech, Manchester, UK), anti-Bcl2 (1:2000 dilution; 12789-1-AP; proteintech, Manchester, UK), anti-ubiquitin (1:1000 dilution; YM3636; Immunoway, Plano, TX, USA) and β-actin (1:5000 dilution; YM3028; Immunoway, Plano, TX, USA). The membranes were then washed in TBST and incubated with secondary antibodies (1:1000 dilution; Invitrogen, Grand Island, NY, USA) for 1 h at room temperature. The blots were scanned using an Odyssey Infrared Imaging System (LI-COR Biosciences, Lincoln, NE, USA). The quantification of antigen–antibody complexes was performed using Quantity One analysis software (Bio-Rad, Hercules, CA, USA).

### 4.6. RT-qPCR Analysis

RNA isolation and RT-PCR were performed as previously described [81]. Briefly, the same amount (15 mg) of each hepatic sample was homogenized in liquid nitrogen, and total RNA was isolated using the miRNeasy mini kit (Qiagen, Hilden, Germany). The same method was used on immortalized HSCs to isolate total RNA. Complementary DNA (cDNA) synthesis was performed using reverse transcription with a PrimeScript RT reagent kit (TaKaRa, Dalian, China). Differences in gene expression were determined using qRT-PCR. The polymerase chain reaction quantification of each sample was performed in triplicate, and SYBR Green fluorescence (TaKaRa) was quantified using the CFX96 real-time system instrument (Bio-Rad, Hercules, CA, USA). The primer sequences for this study are listed in Table 2. Relative expression levels were calculated using the 2^−ΔΔCt^ method, with β-actin as the reference.

### 4.7. IGF-1, IGF-1R, IGFBP-1, and IGFBP-3 Measurement in the Liver, Plasma and Cell Supernatant

Pig liver tissue of the same quality was weighed, anhydrous ethanol was added, the tissue was fully ground, and the supernatant was collected after 3000 r/min centrifugation. Blood samples were collected in tubes containing sodium heparin and centrifuged to obtain plasma fractions. Hepatic and serous IGF-1, IGF-1R, IGFBP-1, and IGFBP-3 levels were measured using the relevant kits according to the manufacturer’s instructions (Sichuan Maccura Biotechnology, Chengdu, China). Hepatic and serous IGF-1, IGF-1R, IGFBP-1, and IGFBP-3 measurements were performed using an automated chemistry analyzer (Hitachi 7600; Hitachi, Tokyo, Japan). Plasma levels of IGF-1, IGF-1R, IGFBP-1, and IGFBP-3 were measured using relevant kits according to the manufacturer’s instructions (Sichuan Maccura Biotechnology, Chengdu, China). Levels of IGF-1, IGF-1R, IGFBP-1, and IGFBP-3 were measured using an automated chemistry analyzer (Hitachi 7600; Hitachi, Tokyo, Japan). Next, the concentrations of IGF-1, IGF-1R, IGFBP-1, and IGFBP-3 in immortalized HSCs supernatants were quantified, using the relevant kits according to the manufacturer’s instructions (Sichuan Maccura Biotechnology, Chengdu, China).

### 4.8. Cell Viability Measurement

The cell suspension was prepared and a blood cell counting plate was used to count. The cell concentration was adjusted to 50,000 cells/mL in the complete medium (10% FBS + ICELL HSCs basal medium + 1% dual antibody + 1% growth factor). Each group had 4 multiple holes, and each hole was inoculated with 100 μL cell suspension (5000 cells) to 96-well plate. This was cultured overnight at 37 °C in a 5% CO_2_ incubator. Virus infection occurred when cell convergence reached 50%. Serum-free and non-double antibody ICELL HSCs basal medium was used to dilute the virus (1:100), and the final concentration of Polybrene was 5 μg/mL (10 μL virus + 0.5 μL 10 mg/mL Polybrene + 1 mL ICELL HSCs basal medium). We changed the 100 μL virus diluent in each well according to the experimental groups. The blank control was not mixed with the virus. This was cultured at 37 °C in a 5% CO_2_ incubator. After overnight culturing, the culture medium (10% FBS + ICELL HSCs basal medium + 1% double antibody + 1% growth factor) was kept at 100 μL at 37 °C and 5% CO_2_. After 72 h of culturing, the corresponding blank medium or 100 μL of the medium containing tunicamycin (5 μg/mL) was replaced according to the experimental groups. This was cultured at 37 °C in a 5% CO_2_ incubator. Tunicamycin was treated for 24 h and 48 h, then the culture medium was sucked out, and the cells were washed once by preheating the serum-free ICELL HSCs basal medium at 37 °C. Added 100 μL the CCK8 working solution (CCK8: ICELL HSCs basal medium = 1:10) to each well and then cultured for 2 h. The A450 was determined using a microplate reader (BioTek, Montpelier, VT, USA).

### 4.9. Statistical Analysis

All statistical analyses were performed using SAS version 9.4 (SAS Institute Inc., Cary, NC, USA). Differences between the means in animal experiment were assessed using one-way analysis of variance (ANOVA), followed by Duncan’s test for multiple comparisons. Differences between the means in the cell experiment containing GRP94 knockdown were assessed using two-way analysis of variance (ANOVA), followed by Bonferroni’s multiple comparison post hoc test. Data are presented as mean ± SD, and statistical significance was set at *p* < 0.05.

## 5. Conclusions

In this study, we used TM to establish the ERS model on piglet livers and immortalized porcine HSCs, which was verified by the significantly increased ERS marker genes and proteins. We found that TM-induced ERS caused an increase in IGF-1 system secretion, and significant ubiquitination without significant apoptosis. On HSCs, under ERS, GRP94 knockdown will influence the IGF system and ERAD process by inhibiting the most conserved UPR sensor protein IRE1α with a significant increase in cell apoptosis. This study deepens our understanding of ERS and supports the molecular basis for targeting GRP94 in ERS-related liver diseases treatment.

## Figures and Tables

**Figure 1 ijms-23-14059-f001:**
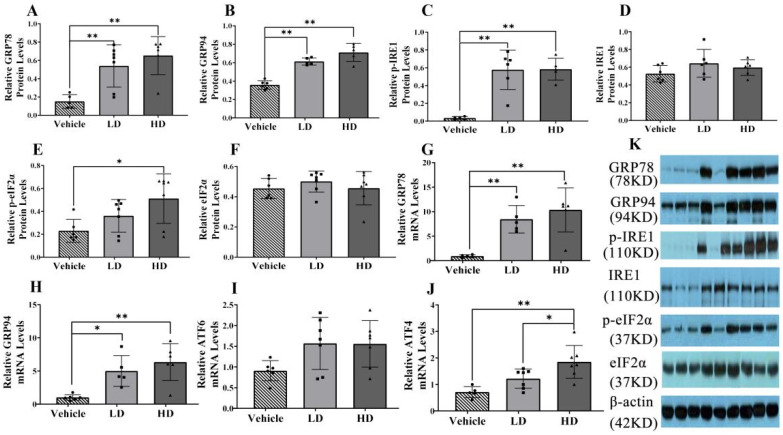
ERS model was successfully established in piglets using different doses of tunicamycin. Piglets were administrated with low-dosage (LD) TM (0.1 mg/kg body weight), high-dosage (HD) TM (0.3 mg/kg body weight), or vehicle (5% DMSO) for 48 h. (**A**–**F**): Relative protein levels of GRP94, GRP78, p-IRE1, IRE1, p-eIF2α, eIF2α, and β-actin protein levels were assessed using Western blot analysis in different TM treatment groups, the relative protein levels of p-IRE1 and p-eIF2α were also obtained by comparing the grey values of their bands and β-actin band. (**G**–**J**): Relative mRNA levels of GRP94, GRP78, ATF6, and ATF4 were measured by real-time RT-PCR. (**K**): Western blot bands. * means *p* < 0.05; ** means *p* < 0.01.

**Figure 2 ijms-23-14059-f002:**
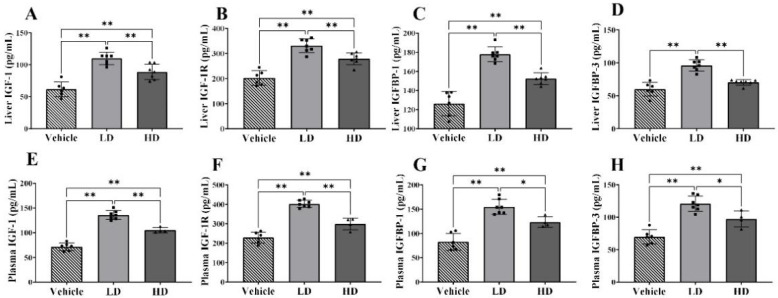
TM-induced ERS effects on IGF-1, IGF-1R, IGFBP-1 and IGFBP-3 in liver and plasma of piglets. Piglets were administrated with TM LD (0.1 mg/kg body weight), HD (0.3 mg/kg body weight), or vehicle for 48 h. (**A**–**D**): Hepatic levels of IGF-1, IGF-1R, IGFBP-1 and IGFBP-3 in different groups. (**E**–**H**): Plasma levels of IGF-1, IGF-1R, IGFBP-1 and IGFBP-3 in different groups. * means *p* < 0.05; ** means *p* < 0.01.

**Figure 3 ijms-23-14059-f003:**
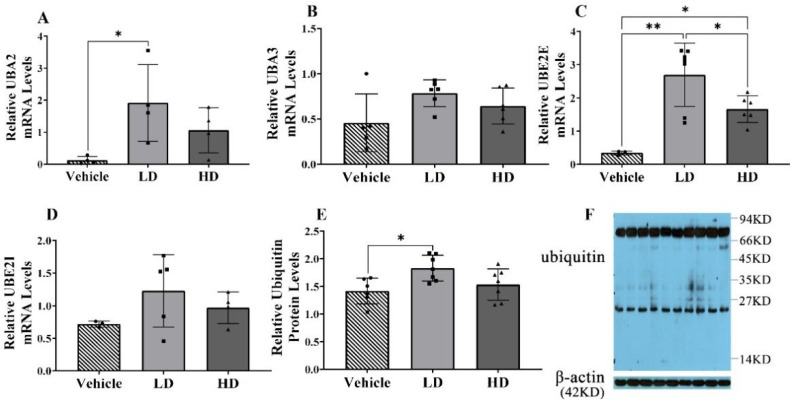
TM-induced ERS effects on ubiquitin-related genes and ubiquitin-protein in piglets. Piglets were administrated with TM LD (0.1 mg/kg body weight), HD (0.3 mg/kg body weight), or vehicle for 48 h. (**A**–**D**): Relative mRNA levels of UBA2, UBA3, UBE2E, and UBE2I were assessed using real-time RT-PCR in different groups. (**E**): Relative protein levels of ubiquitin and β-actin were assessed using Western blot. (**F**): Western blot bands. * means *p* < 0.05; ** means *p* < 0.01.

**Figure 4 ijms-23-14059-f004:**
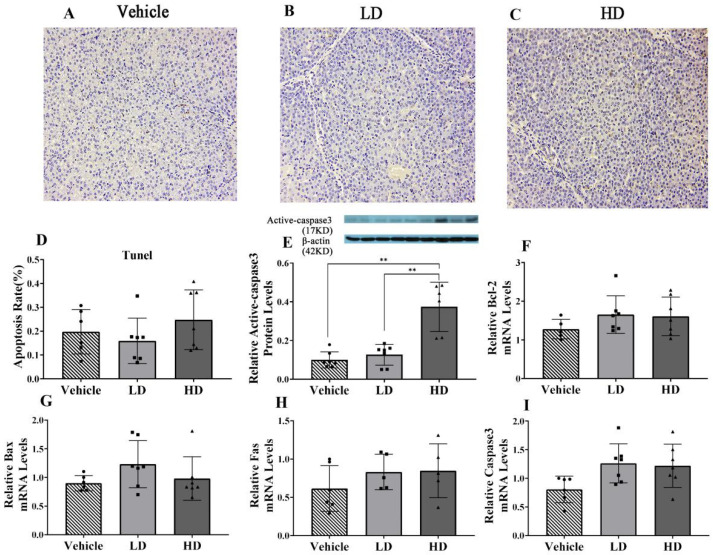
TM-induced ERS effects on liver cell apoptosis in piglets. Piglets were administrated with TM LD (0.1 mg/kg body weight), HD (0.3 mg/kg body weight), or vehicle for 48 h. (**A**–**C**): TUNEL staining of vehicle, LD, and HD groups under 200× objective lens. (**D**): The apoptosis rate. (**E**): Relative protein levels of Active-Caspase 3 and β-actin protein levels were assessed using Western blot analysis in different TM treatment groups. (**F**–**I**): Relative mRNA levels of GRP94, GRP78, ATF6, and ATF4 were measured by real-time RT-PCR. ** means *p* < 0.01.

**Figure 5 ijms-23-14059-f005:**
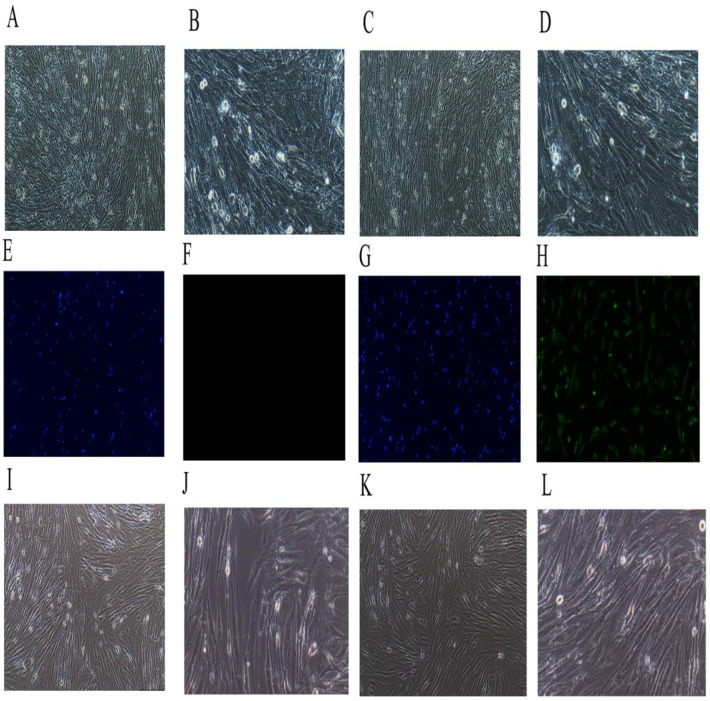
Primary porcine hepatic stellate cells’ (HSCs) immortalization. (**A**,**B**): The separated porcine HSCs; (**A**) the observation result under a 100× objective lens, and (**B**) the observation result under a 200× objective lens. (**C**,**D**): The separated primary porcine HSCs cultured for 3–5 d; (**C**) the observation result under a 100× objective lens, and (**D**) the observation result under a 200× objective lens. (**E**,**F**): The immunofluorescence map of the control group; (**E**) the immunofluorescence map of 100×–DAPI, and (**F**) the immunofluorescence map of 100×–fluorescence. (**G**,**H**): The immunofluorescence map of the test group; (**G**) the immunofluorescence map of 100×–DAPI, and (**H**) the immunofluorescence map of 100×–fluorescence. (**I**,**J**): The successfully transfected porcine hepatic stellate cells of the first generation (Pl) were obtained after the screening; (**I**) the map observed under a 100× objective lens, and (**J**) the map observed under a 200× objective lens. (**K**,**L**): The successfully transfected porcine hepatic stellate cells of the first generation (Pl) were passaged until the tenth generation (P10); (**K**) the map observed under a 100× objective lens, and (**L**) the map observed under a 200× objective lens.

**Figure 6 ijms-23-14059-f006:**
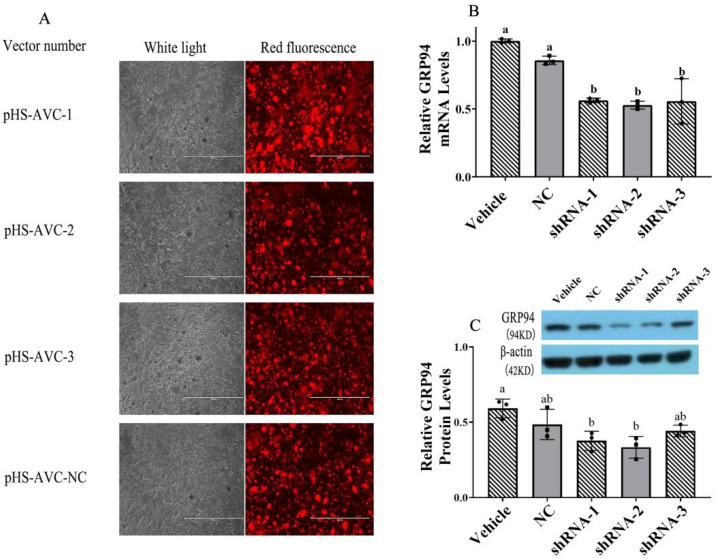
GRP94 shRNA effects on GRP94 gene and protein expressions. (**A**): The fluorescence photograph of pHS-AVC-NC (negative control (NC)), pHS-AVC-1 (GRP94-shRNA1), pHS-AVC-2 (GRP94-shRNA2) and pHS-AVC-3 (GRP94-shRNA3) observed under a 100× objective lens; the virus titer was calculated by the expression of the fluorescence. (**B**,**C**): The effects of shRNA on the expression of the GRP94 protein and gene were measured using real-time RT-PCR and Western blot. The same letters mean there was no significant difference (*p* < 0.05) between the groups.

**Figure 7 ijms-23-14059-f007:**
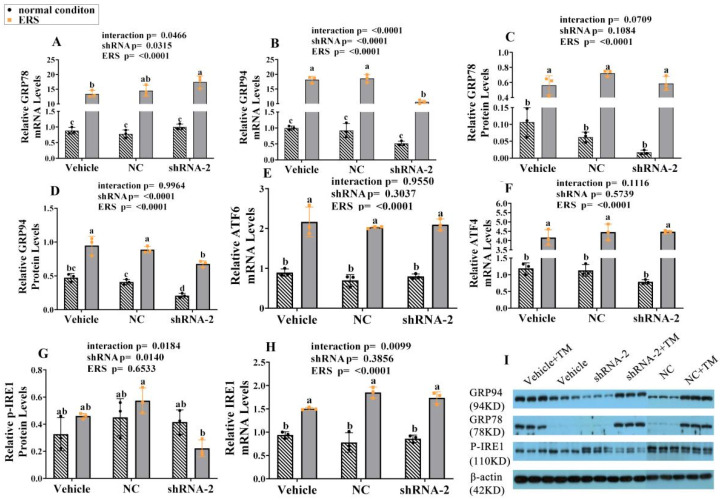
GRP94 knockdown effects on ERS-related genes and proteins in immortalized porcine HSCs under ERS and normal condition. (**A**,**B**,**E**,**F**,**H**): Relative mRNA levels of GRP78, GRP94, ATF6, ATF4 and IRE1 were measured using real-time RT-PCR; (**C**,**D**,**G**): Relative protein levels of GRP78, GRP94 and p-IRE1 were assessed using Western blot, and the relative protein levels of p-IRE1 were also obtained through comparing the grey values of p-IRE1 bands and β-actin band. (**I**): Western blot bands. Data were assessed with two-way ANOVA (main effects: ERS, shRNA, and their interaction), followed by Bonferroni’s multiple comparison post hoc test. The same letters mean no significant difference (*p* < 0.05) between groups.

**Figure 8 ijms-23-14059-f008:**
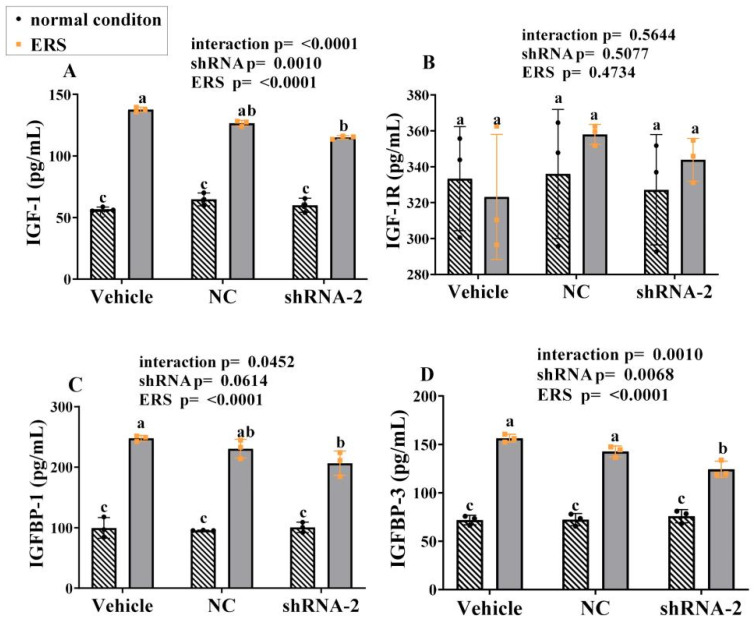
Effects of knockdown GRP94 on the expression of IGF-1, IGF-1R, IGFBP-1, and IGFBP-3 in the cell supernatant of immortalized porcine HSCs. (**A**–**D**): The levels of IGF-1, IGF-1R, IGFBP-1 and IGFBP-3 in the cell supernatant of immortalized HSCs. Data were assessed with two-way ANOVA (main effects: ERS, shRNA, and their interaction), followed by Bonferroni’s multiple comparison post hoc test; the same letters mean no significant difference (*p* < 0.05) between groups.

**Figure 9 ijms-23-14059-f009:**
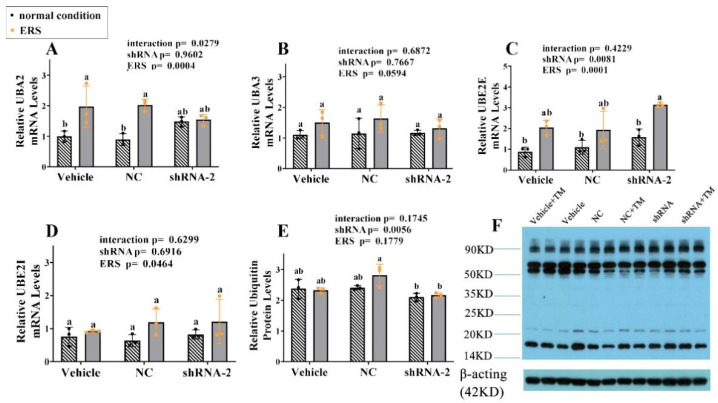
Effects of GRP94 knockdown on ubiquitin-related genes and ubiquitin proteins in immortalized porcine HSCs. (**A**–**D**): Relative mRNA levels of UBA2, UBA3, UBE2E and UBE2I in immortalized HSCs were measured using real-time RT-PCR. (**E**): Relative protein levels of ubiquitin in immortalized HSCs were assessed using Western blot. (**F**): Western blot bands. Data were assessed with two-way ANOVA (main effects: ERS, shRNA, and their interaction), followed by Bonferroni’s multiple comparison post hoc test; the same letters mean no significant difference (*p* < 0.05) between groups.

**Figure 10 ijms-23-14059-f010:**
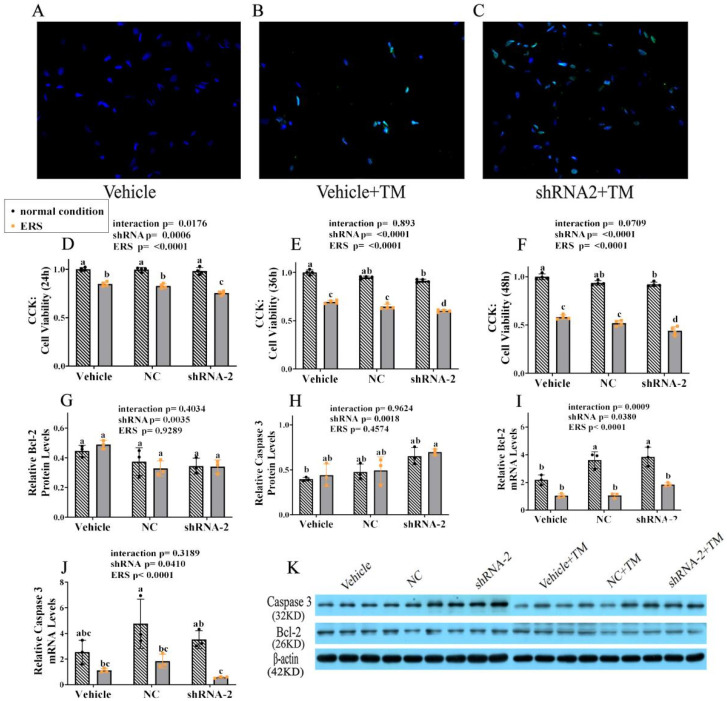
GRP94 knockdown effects on immortalized porcine HSCs apoptosis. (**A**–**C**): TUNEL staining of immortalized HSCs observed under a 200× objective lens. (**D**–**F**): Cell proliferation assay of immortalized HSCs after TM treatment for 24 h, 36 h and 48 h by CCK-8 assay. (**G**,**H**): Relative protein levels of Bcl-2 and Caspase 3 in immortalized HSCs were assessed using a Western blot. (**K**): Western blot band. (**I**,**J**): Relative mRNA levels of Bcl-2 and Caspase 3 in immortalized HSCs were measured using real-time RT-PCR. Data were assessed with two-way ANOVA (main effects: ERS, shRNA, and their interaction), followed by Bonferroni’s multiple comparison post hoc test; the same letters mean no significant difference (*p* < 0.05) between groups.

**Figure 11 ijms-23-14059-f011:**
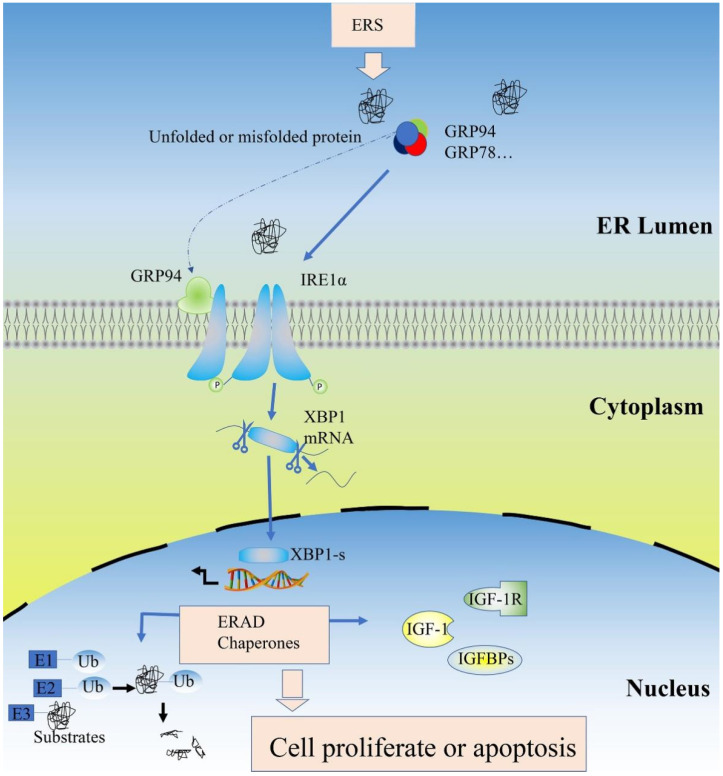
Schematic of the GRP94 pathway under ERS. ERS: endoplasmic reticulum stress; GRP78: glucose regulated protein 78; GRP94: glucose-regulated protein 94; IRE1α: inositol requiring enzyme 1; XBP1: X-box-binding protein 1; XBP1-s: spliced X-box-binding protein 1; ERAD: endoplasmic reticulum-associated degradation; IGF-1: insulin-like growth factor-1; IGF-1R: insulin-like growth factor-1 receptor; IGFBPs: insulin-like growth factor-binding proteins; Ub: ubiquitin; E1: ubiquitin-activating enzymes; E2: ubiquitin conjugating enzymes; E3: ubiquitin–protein ligase.

**Table 1 ijms-23-14059-t001:** Gene-silencing lentivirus vector information.

Carrier No.	Insert Content	shRNA Sequence
pHS-AVC-1	GRP94-shRNA1	5′-GCTGGAAATGAGGAGTTAACG CGAACGTTAACTCCTCATTTCCAGC-3′
pHS-AVC-2	GRP94-shRNA2	5′-GCAGCAACATAAACTCCTTAA CGAATTAAGGAGTTTATGTTGCTGC-3′
pHS-AVC-3	GRP94-shRNA3	5′-GGAAAGAATTTGGTACCAACA CGAATGTTGGTACCAAATTCTTTCC-3′

**Table 2 ijms-23-14059-t002:** Specific primer sequences used in quantitative real-time PCR.

Gene Name	Primer	Product (bp)
GRP94	F: ACACTGCGGTCAGGGTAT R: TTCTTCGTCGTCGTCTTG	180
GRP78	F: AATGGCCGTGTGGAGATCA R: GAGCTGGTTCTTGGCTGCAT	114
ATF6	F: CCAGTCCTTGCTGTCACT R: CACACTTCTCATGGTACTCTG	141
ATF4	F: GCCCCCGCAGATAGTGAA R: TGGGGAAAGGGGAAGAGTTTG	103
UBA2	F: CAGTGCAAAAAGGTCACGCA R: GGCCTAATTCGGACAAGGGT	105
UBA3	F: GCGGAGAACAATATGGCGGA R: ACCTTCCCAGTCTCCAGTGT	120
UBE2E	F: CATGCTCACCAAGTGCATCG R: GTGAAGGGTTGGGAAGGGAG	136
UBE2I	F: ATTCCACCCGAACGTGTACC R: ATCGTGTAGGCCTCTGCTTG	165
Bcl2	F: TGGAGAGCGTAGACAAGGAGA R: CATCGGTTGAAGCGTTCCTG	184
Caspase-3	F: AGAATTGGACTGTGGGATTGAGACG R: GCCAGGAATAGTAACCAGGTGCTG	122
Bax	F: GCTGACGGCAACTTCAACTG R: CCGATCTCGAAGGAAGTCCA	141
Fas	F: TGATGCCCAAGTGACTGACC R: GCAGAATTGACCCTCACGAT	161

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
