# Peer review of "GRP94 Inhabits the Immortalized Porcine Hepatic Stellate Cells Apoptosis under Endoplasmic Reticulum Stress through Modulating the Expression of IGF-1 and Ubiquitin"

_ijms, 2022, doi:10.3390/ijms232214059_

Round 1

Reviewer 1 Report

General Comment:

The Authors demonstrated GRP94 can protect the cell from ERS-induced apoptosis by promoting IGF-1 system and ubiquitin. This study is an interesting and timely study exploring adaptive mechanisms of the liver under ERS and could help identify vital functional genes to apply as possible diagnostic biomarkers and treatments for diseases induced by ERS in the future. Overall, this manuscript is well written and up-to-date. However, it requires some improvements.

My minor criticisms relate to the following issues:

1.                   All figures require adjustment with the fonts. Currently are too small and hard to read them.

2.                   Space between words and references in the text requires to be adjusted.

3.                   Fig.1 G requires a line under HD. In addition, to improve the confidence in the accuracy of western blot results Authors should describe in figure legend/methods how they normalize phosphorylated proteins analyzed by Western blot in order to quantify the levels of P-IRE1 and phosphorylated EIF2α.  

4.                   Pictures in panels Fig.5, 6A and 7I, 9F are poor quality and need to be improved.

Author Response

Dear Reviewer:

Thank you for your letter and for your comments concerning our manuscript entitled “GRP94 inhabits the immortalized porcine hepatic stellate cells apoptosis under endoplasmic reticulum stress through modulating the expression of IGF-1 and ubiquitin”. Those comments are all valuable and very helpful for revising and improving our paper, as well as the important guiding significance to our research. We have studied comments carefully and have made correction which we hope to meet with approval. Revised portions are marked in red in the paper. The main corrections in the paper and the responds to your comments are as follows:

1.All figures require adjustment with the fonts. Currently are too small and hard to read them.

Response: As the reviewer suggested. We have resubmitted the renewed figures in the form of vector diagrams (.svg), in this way the figures will look clearer. Thanks for your noble advice.

2.Space between words and references in the text requires to be adjusted.

Response: As the reviewer suggested. We have adjusted the space between words and references in the text. Thanks for your noble advice.

3.Fig.1 G requires a line under HD. In addition, to improve the confidence in the accuracy of western blot results Authors should describe in figure legend/methods how they normalize phosphorylated proteins analyzed by Western blot in order to quantify the levels of P-IRE1 and phosphorylated EIF2α.

Response: As the reviewer suggested. We have adjusted the layout of figure 1 (line 119), and we added the description that the relative protein levels of phosphorylated proteins were obtained by comparing the grey values of their bands and β-acting band (line 124 and 264), because there is no significant differences founded in the eIF2α and IRE1 under different treatment, so statistically we didn`t compare the phosphorylated protein to the total protein again. Thanks for you noble advise.

4.Pictures in panels Fig.5, 6A and 7I, 9F are poor quality and need to be improved.

Response: As the reviewer suggested. We have resubmitted the renewed figures in the form of vector diagrams (.svg), in this way the figures will look clearer. Thanks for your noble advice.

Reviewer 2 Report

In this study, Wang et al. seek the roles of GRP94 on tunicamycin-induced UPR using immortalized HSCs. Although the idea is tempting, this study lacks clear rationale and critical evidence to support the authors’ conclusion.

·       The rationale to use porcine models, tunicamycin, and HSCs is unclear. The authors say that piglets share more similarities to human, but administration of tunicamycin does not mimic conditions of human patients at all. The authors mention NAFLD and NASH in Introduction, but tunicamycin treatment in pigs is not a model of NAFLD or NASH. Rodents with high-fat diet or MCD diet etc. will be better models for NAFLD or NASH. Tunicamycin-treated pigs are a model of what disorder? It is not science to treat animals with random chemicals and claim protein/gene expression differences and mechanisms. The authors need to describe clearly why they use pigs with tunicamycin. In addition, the authors use immortalized HSCs, but the rationale is unclear. The authors say “HSCs as lipid storage cell. However, hepatocytes consist about 70% of liver cell population and lipid droplets are seen mainly in hepatocytes in NAFLD. The authors also say “HSCs will transform from quiescent state to activation under stress, so HSCs are suitable for ERS study”. However, this is clearly wrong because many previous studies showed that HSCs are activated by TGF-β1, not UPR, leading to liver fibrosis. A study shows that UPR is not a critical event for HSC activation (Mannaerts et al., Cell Death & Disease, 2019). There is no reason to immortalize HSCs as a model of UPR. Furthermore, HSCs are closely associated with hepatic fibrogenesis, but the authors ignore HSC functions and merely study apoptosis. What disease model is the treatment of HSCs with tunicamycin? NAFLD or acute liver failure or cholangiopathies or what? Accumulation of ERS in HSCs leads to liver fibrosis? In which disorder? This study lacks critical rationale and aims of study.

·       Tunicamycin administration did not induce apoptosis (Figure 4), which is strange. Tunicamycin causes apoptosis via UPR accumulation, and that is why some studies show that tunicamycin may be an anti-cancer drug. Tunicamycin treatment was insufficient or something was wrong with procedures if the authors saw UPR but not apoptosis. Characterization of animals is poorly done and data are limited. ALT/AST should be shown and H&E staining, Sirius Red, and Oil-Red-O staining should also be included because the authors look at HSC activation and tunicamycin induces lipid accumulation in the liver (Kim et al. Int J Mol Sci, 2018). It is unclear if this animal model is suitable (for which disorder? Or simply UPR model?) or data the authors show here are merely random accidents.

·       Western images are inconsistent so not reliable. In Figure 6C, HSCs with vehicle (first lane) and negative control (second lane) show medium-strong bands for GRP94. However, in Figure 7I, NC+normal and Vehicle+normal groups clearly show weaker bands for GRP94 compared to those in Figure 6C. Even in Figure 7I, Vehicle+normal looks stronger for GRP94 than NC+normal. These cells are control cells and should show similar bands. GRP94 for vehicle or normal control in Figure 6C looks similar to bands in NC+TM in Figure 7I. It is impossible to believe the authors’ claims if control group shows similar intensity in one image to tunicamycin group in the other image. Same problem for ubiquitin bands as well. This inconsistency raises a serious question that immortalized HSCs are not stable or inconsistent or contaminated, so the authors obtained different results in different experiments. It seems that data are not repeatable and not reliable.

Author Response

Dear Reviewer:

Thank you for your letter and for your comments concerning our manuscript entitled “GRP94 inhabits the immortalized porcine hepatic stellate cells apoptosis under endoplasmic reticulum stress through modulating the expression of IGF-1 and ubiquitin”. Those comments are all valuable and very helpful for revising and improving our paper, as well as the important guiding significance to our research. We have studied comments carefully and have made correction which we hope to meet with approval. Revised portions are marked in red in the paper. The main corrections in the paper and the responds to your comments are as follows:

Response: Thanks for your noble advice. About ERS model. ERS is one of reasons leading to the occurrence and deterioration of many liver diseases, so we used the tunicamycin to establish ERS model on pigs to mimic the early phase of liver disease. We also used more ink in the text to re-elaborate the relationship between ERS and the occurrence of liver disease and apoptosis (line 40). Pig (Sus scrofa) is emerging as an attractive biomedical model for studying liver diseases in humans because of their similar metabolic features function[1]. So our results in this study can give more valuable information about how people react ERS than studies conducted on mice or rats. There is no doubt that if we want to establish a NAFLD or NASH model, we should use HFD or MCD diet model. In this study, we used TM to establish an acute ERS model to mimic the acute liver disease like acute liver failure (ALF) instead of just metabolic liver disease. There are studies supported that the close relationship between ERS and ALF[2]. What`s more regulating the ERS process can even improve ALF[3].

Response: About HSCs. Mannaerts et al mentioned that “A timely termination of the unfolded protein response is essential to prevent endoplasmic reticulum stress-related apoptosis.” So in his point of view, UPR is a compensatory mechanism to prevent disastrous ER stress levels which could lead to cell death. But in fact, ERS can lead to two results on cell apoptosis. Although slight ERS cannot induce cell apoptosis, serious and prolonged ERS can induce cell apoptosis. In Mannaerts`s study, he used 0.5 μg/ml to treat cells. We think the concentration is not sufficient enough, which may explain why he just observed UPR during a very limited period in early phase. In our study, we used 5 μg/ml TM to treat cells, which is closer to the acute ERS condition. In our study, we indeed observed the activation of UPR on HSCs, and the CCK-8 results also proved the constant inhabitation of ERS on cell viability. What`s more, there are many studies used the HSCs to explore the connection between the ERS induced by TM and apoptosis[4-6], liver diseases like fibrosis[7] and acute liver failure[8] and so on. So we think it`s suitable to use HSCs as cell materials for ERS study.

Response: About the dose of TM in our experiment. There are studies proved that the destructive degree of ER structure is depended on the dose of the drug administrated[9] and there is a dose-dependent apoptosis induction by TM[10]. So we don`t think there was something wrong with procedures. The dose of TM is obtained from the body surface area-body weight conversion formula, so we have set low-dose group and high-dose groups. Additionally, TM-treated piglets have appeared serious pathological phenomena such as steatosis of liver and significantly decreased feed intake, so we think that the dose is appropriate. And we think the results of no apoptosis under ERS are also understandable, because the body has a much more regulated ability than cells. We do observe the significant ERS on piglet livers, thuswe take more attention to what specific ways the body regulates ERS.

Furthermore, this study focuses more on hepatic cell apoptosis, rather than lipid metabolism, so we don`t list the results of Oil-Red-O staining. The data about HE staining plot, ALT/AST and lipid metabolism have already been published in frontiers in physiology[11], these data showed that TM treatment on piglets led to liver steatosis and liver injury.

Special thanks to you for your good comments.

We tried our best to improve the manuscript and made some changes in the manuscript. These changes will not influence the content and framework of the paper. And here we did not list the changes but marked in red in revised paper. We appreciate for your warm work earnestly, and hope that the revised manuscript is now suitable for publication in your esteemed journal.

Reviewer 3 Report

The authors studied the relationship between GRP94 as endoplasmic reticulum (ER) chaperone and the occurrence and progression of metabolic liver disease, and verified that GRP94 can protect the hepatocyte from endoplasmic reticulum stress (ERS)-induced apoptosis by promoting IGF-1 system and ubiquitin. The manuscript is generally well presented with the logical thinking and structure. Demonstrated the value of GRP94 in liver diseases to a certain extent. There are a few aspects the authors could consider further as outlined below:

Recommendations:

1.     The article experimentally proved that the changes of apoptosis-related factors caused by GRP94 knockdown did not significantly interact with ERS, indicating that GRP94 plays an important role in cell fate. Under normal circumstances, why can it be proved that GRP94 must also play an important role in ERS? Please explain.

2.     In this paper, in addition to GRP94, GRP78 is also mentioned, both in the introduction (lines 68-69), the experiments and the discussion. Especially in the discussion comparing the two ER chaperone, showing the advantages of GRP78, why not use knockdown GRP78 for experimental exploration?

3.     Please check the text format, for examples:

      i.           The first letter of the subtitle also needs to be capitalized, in lines 100 and 175, “establishment” should be “Establishment”; “immortalized” should be “Immortalized”.

     ii.           In lines 215 and 217, delete the underline of the “fluorescence”.

   iii.           In line 377, the “20 fold” should be “20-fold”, same as your reference [46].

4.     In the results of Western blot, the molecular weight size of the protein is preferably marked in the figure.

5.     GAPDH as housekeeping protein appears in figure 3F and is not described in the methods. At the same time, β-actin is used in other experiments, please use only one housekeeping protein for checking equal loading of each sample. Or please explain why two different housekeeping proteins were used.

6.     There are some details in the picture that need to be adjusted, such as:

      i.           In figure 6A, the scale in the fluorescence photograph cannot be seen clearly.

     ii.           In figure 1G, HD is missing a horizontal line.

   iii.           The legends in many figures are not clear, especially in figures 7-10. It is recommended to increase the clarity of all figures.

   iv.           In figure 7D, the legend “94 protein” should be “GRP94 protein”.

7.     Please explain figure 9B in “2.9 The effect of GRP94 knockdown and ERS on ubiquitination in HSCs”, whether there is a significant difference in UBA3, which is not introduced.

8.     Since the focus of this manuscript is to prove the efficacy of GRP94 in apoptosis, whether more relevant experiments are needed to further verify and make the data more convincing. In the results "2.10. The effect of GRP94 knockdown and ERS on apoptosis in HSCs", in addition to using Western blot to detect the expressions of proteins, the expressions of apoptosis-related genes can also be detected.

9.     XBP1 is the most conserved UPR branch, in lines 355-360, introduced that “GRP94 deletion leads to the decrease of the cleavage of XBP1 protein of IRE1”, but the experiment did not study the protein level of XBP1, only the expression of IRE1 protein, how to prove that it must be effect of the XBP1 pathway.

10.  Note that the piglet models, TM was dissolved in DMSO. As we all know, DMSO has certain toxicity, is there any toxicity test to exclude the influence of DMSO?

11.  In the Materials and Methods, the subtitles of “4.7” and “4.8” are the same. The subtitle of “4.8” needs to be revised, which is obviously inconsistent with the content of the text.

12.  Please indicate the phosphorylation sites of certain proteins in the manuscript.

Author Response

Dear Reviewer:

Thank you for your letter and for your comments concerning our manuscript entitled “GRP94 inhabits the immortalized porcine hepatic stellate cells apoptosis under endoplasmic reticulum stress through modulating the expression of IGF-1 and ubiquitin”. Those comments are all valuable and very helpful for revising and improving our paper, as well as the important guiding significance to our research. We have studied comments carefully and have made correction which we hope to meet with approval. Revised portions are marked in red in the paper. The main corrections in the paper and the responds to your comments are as follows:

1.The article experimentally proved that the changes of apoptosis-related factors caused by GRP94 knockdown did not significantly interact with ERS, indicating that GRP94 plays an important role in cell fate. Under normal circumstances, why can it be proved that GRP94 must also play an important role in ERS? Please explain.

Response: Thanks for your noble advice. The lack of interaction between ERS and shRNA on the change of apoptosis factors indicates the GRP94 approximately has the same anti-apoptosis effect in both ERS and normal conditions. What`s more the GRP94 will expressed highly under ERS, the change of expression is the premise that the molecule has function. GRP94 also showed the anti-apoptosis effect similar to those under normal condition, which indicated that the susceptibility of cell to the expression of GRP94 under ERS.

2.In this paper, in addition to GRP94, GRP78 is also mentioned, both in the introduction (lines 68-69), the experiments and the discussion. Especially in the discussion comparing the two ER chaperone, showing the advantages of GRP78, why not use knockdown GRP78 for experimental exploration?

Response: Thanks for your noble advice. GRP78 is a marked protein of ERS which already has been well studied in terms of how to exert anti-apoptosis effects under ERS. GRP78 has been proved that take part in the activation of IRE1, ATF6 and PERK pathway. On the other hand, GRP94 is also expressed highly as a glucose regulatory protein, and the specific effect of it on apoptosis and by which pathways to effect are still controversial. In addition, the number of binding proteins of GRP94 is less than that of GRP78. So it`s of great sense to explore the agent proteins of GRP94. Based on the evidence above, we chose GRP94 to study.

3.Please check the text format, for examples:

i.The first letter of the subtitle also needs to be capitalized, in lines 100 and 175, “establishment” should be “Establishment”; “immortalized” should be “Immortalized”.

Response: As the reviewer suggested. We have corrected the relative subtitle.

ii.In lines 215 and 217, delete the underline of the “fluorescence”.

Response: As the reviewer suggested. We have deleted the underline.

iii. In line 377, the “20 fold” should be “20-fold”, same as your reference [46].

Response: As the reviewer suggested. We have changed the “20 fold” to “20-fold.”

4.In the results of Western blot, the molecular weight size of the protein is preferably marked in the figure.

Response: Thanks for your noble advice. We have marked the molecular weight size of the protein in figure.

5.GAPDH as housekeeping protein appears in figure 3F and is not described in the methods. At the same time, β-actin is used in other experiments, please use only one housekeeping protein for checking equal loading of each sample. Or please explain why two different housekeeping proteins were used.

Response: Thanks for your noble advice. Because the materials of this experiment are both mammalian tissue or mammalian cells, both GAPDH and β-actin can be used as housekeeping proteins. So in some WB experiments, we used two housekeeping proteins, and we have changed all the blots to β-actin blots.

6.There are some details in the picture that need to be adjusted, such as:

i.In figure 6A, the scale in the fluorescence photograph cannot be seen clearly.

Response: As the reviewer suggested. We have resubmitted the renewed figures in the form of vector diagrams (.svg), in this way the figures will look clearer. Thanks for your noble advice.

ii.In figure 1G, HD is missing a horizontal line.

Response: As the reviewer suggested. We have adjusted the typography of figure1 to make the figure look clearer. Thanks for your noble advice.

iii. The legends in many figures are not clear, especially in figures 7-10. It is recommended to increase the clarity of all figures.

Response: As the reviewer suggested. We have resubmitted the renewed figures with more obvious legends in figures 7-10. Thanks for your noble advice.

iv.In figure 7D, the legend “94 protein” should be “GRP94 protein”.

Response: As the reviewer suggested. We have corrected the “94 protein” to “GRP94 protein”. Thanks for your noble advice.

7.Please explain figure 9B in “2.9 The effect of GRP94 knockdown and ERS on ubiquitination in HSCs”, whether there is a significant difference in UBA3, which is not introduced.

Response: Thanks for your noble advice. Maybe because the previous figure has not enough clarity, statistically there is no significant difference found in UBA3 expressions, shRNA even has little effect on its expressions from the aspect of two-way ANOVA.

8.Since the focus of this manuscript is to prove the efficacy of GRP94 in apoptosis, whether more relevant experiments are needed to further verify and make the data more convincing. In the results "2.10. The effect of GRP94 knockdown and ERS on apoptosis in HSCs", in addition to using Western blot to detect the expressions of proteins, the expressions of apoptosis-related genes can also be detected.

Response: As the reviewer suggested. The gene changes of apoptosis factors are also needed to explore, so we conducted the qPCR experiments, and added the relative results on figure 10. The changes of genes and proteins are not consistent, the qPCR results showed that ERS decreased both the Caspase 3 and increased the Bcl-2 mRNA expressions. In the aspect of shRNA, it seems that there is no significant change trend. In the point of view, the decreased Caspase 3 mRNA expressions under ERS indicates there is a negative feedback regulation of HSCs to prevent the constant cell apoptosis leaded by ERS. In addition, protein is the basic unit of executive function in organism, so we take more attention on the changes of proteins to explain the condition of apoptosis. We also adjusted the content of manuscript to explain the qPCR results. Thanks for your noble advice.

9.XBP1 is the most conserved UPR branch, in lines 355-360, introduced that “GRP94 deletion leads to the decrease of the cleavage of XBP1 protein of IRE1”, but the experiment did not study the protein level of XBP1, only the expression of IRE1 protein, how to prove that it must be effect of the XBP1 pathway.

Response: Thanks for your noble advice. “GRP94 deletion leads to the decrease of the cleavage of XBP1 protein of IRE1” is the summarize of a reference[12], which proved that GRP94 can affect the XBP-1. And the relationship between IRE1 and XBP1 is internationally recognized. So the results of this reference supports our experiments of there is a close relationship between GRP94 and IRE-1.

10.Note that the piglet models, TM was dissolved in DMSO. As we all know, DMSO has certain toxicity, is there any toxicity test to exclude the influence of DMSO?

Response: Thanks for your noble advice. It`s widely recognized that despite some growth inhibition caused by DMSO, the concentration range of 5–10% v/v was found appropriate for bringing sparingly water-soluble chemicals into aquatic solutions[13]. In the pre-experimental stage, we set up the DMSO control group, and it was not found that DMSO had a significant effect on the physiological status of piglets. So we think the concentration of DMSO is appropriate.

11.In the Materials and Methods, the subtitles of “4.7” and “4.8” are the same. The subtitle of “4.8” needs to be revised, which is obviously inconsistent with the content of the text.

Response: As the reviewer suggested. We have corrected the subtitle of “4.8” to “cell viability measurement”. Thanks for your noble advice.

12.Please indicate the phosphorylation sites of certain proteins in the manuscript.

Response: As the reviewer suggested. The phosphorylated site of IRE1 is at the cytol domain,[14] so we drew the phosphorylated molecule in cytoplasm. Thanks for your noble advice.

Special thanks to you for your good comments.

We tried our best to improve the manuscript and made some changes in the manuscript. These changes will not influence the content and framework of the paper. And here we did not list the changes but marked in red in revised paper. We appreciate for your warm work earnestly, and hope that the revised manuscript is now suitable for publication in your esteemed journal.

Round 2

Reviewer 2 Report

The authors did not address my comments, including concerns for models, effects of tunicamycin, and inconsistent data. I cannot accept the manuscript since it was not improved.

Author Response

Dear review,
We have studied your comments carefully concerning our manuscript entitled “GRP94 inhabits the immortalized porcine hepatic stellate cells apoptosis under endoplasmic reticulum stress through modulating the expression of IGF-1 and ubiquitin”.
In response to the questions raised by you, we believe that we have made the best efforts in the response and the draft.
For the results of WB, we think this color difference is acceptable, within the normal range.
For the TM, you think that administration of TM does not mimic conditions of human patients at all and TM treatment in pigs is not a model of NAFLD or NASH. TM is an internationally recognized ERS inducer, in the response for your question, we have written a lot to prove that there is a link between ERS and the occurrence and development of various liver diseases, not only NAFLD but also acute liver diseases such as acute liver failure[1, 2]. So we think it is appropriate to use TM to simulate acute ERS associated with many liver diseases.
For the animal model, we think the pig (Sus scrofa) is emerging as an attractive biomedical model for studying liver diseases in humans because of their similar metabolic features function[3]. No matter in body size or in metabolic function, pigs are closer to humans than model rats and mice obviously. So our results in this study can give more valuable information about how people react to ERS than studies conducted on mice or rats.
For the cell model, you think that UPR is not a critical event for HSC activation and there is no reason to immortalize HSCs as a model of UPR. It is also controversial that you pointed out that UPR is not an important reason for HSC activation. You reached this conclusion from a reference in which the experimenter used 0.5 to deal with cells. We think the concentration is not sufficient enough, which may explain why he just observed UPR during a very limited period in early phase. In our study, we used 5 μg/ml TM to treat cells, which is closer to the acute ERS condition. And actually 

And actually there are many studies used the HSCs to explore the connection between the ERS induced by TM and cell apoptosis[4-6], liver diseases like fibrosis[7] and acute liver failure[8] and so on. So HSC is a common material for studying ERS.

We have tried our best to improve the manuscript quality and reply to the questions of you.

We appreciate your warm work earnestly, and hope that the revised manuscript is now suitable for publication in your esteemed journal.

  1. Liu, Y.; Pan, X.; Li, S.; Yu, Y.; Chen, J.; Yin, J.; Li, G., Endoplasmic reticulum stress restrains hepatocyte growth factor expression in hepatic stellate cells and rat acute liver failure model. Chem. Biol. Interact. 2017, 277, 43-54.
  2. Torres, S.; Baulies, A.; Insausti-Urkia, N.; Alarcón-Vila, C.; Fucho, R.; Solsona-Vilarrasa, E.; Núñez, S.; Robles, D.; Ribas, V.; Wakefield, L.; Grompe, M.; Lucena, M. I.; Andrade, R. J.; Win, S.; Aung, T. A.; Kaplowitz, N.; García-Ruiz, C.; Fernández-Checa, J. C., Endoplasmic Reticulum Stress-Induced Upregulation of STARD1 Promotes Acetaminophen-Induced Acute Liver Failure. Gastroenterology 2019, 157, (2), 552-568.
  3. Jeong, P. S.; Yoon, S. B.; Lee, M. H.; Son, H. C.; Lee, H. Y.; Lee, S.; Koo, B. S.; Jeong, K. J.; Lee, J. H.; Jin, Y. B.; Song, B. S.; Kim, J. S.; Kim, S. U.; Koo, D. B.; Sim, B. W., Embryo aggregation regulates in vitro stress conditions to promote developmental competence in pigs. PeerJ 2019, 7, e8143.
  4. Liu, H.; Dai, L.; Wang, M.; Feng, F.; Xiao, Y., Tunicamycin Induces Hepatic Stellate Cell Apoptosis Through Calpain-2/Ca(2 +)-Dependent Endoplasmic Reticulum Stress Pathway. Front Cell Dev Biol 2021, 9, 684857.
  5. Huang, Y.; Li, X.; Wang, Y.; Wang, H.; Huang, C.; Li, J., Endoplasmic reticulum stress-induced hepatic stellate cell apoptosis through calcium-mediated JNK/P38 MAPK and Calpain/Caspase-12 pathways. Mol Cell Biochem 2014, 394, (1-2), 1-12.
  6. Zhang, J.; Yang, L.; Han, X.; Li, C.; Liu, R.; Ma, Z.; Han, B.; Xie, R.; Yang, Q., [Endoplasmic reticulum stress in hepatic stellate cells induced by tunicamycin promotes apoptosis and cell cycle arrest]. Xi Bao Yu Fen Zi Mian Yi Xue Za Zhi 2021, 37, (9), 794-800.
  7. De Minicis, S.; Candelaresi, C.; Agostinelli, L.; Taffetani, S.; Saccomanno, S.; Rychlicki, C.; Trozzi, L.; Marzioni, M.; Benedetti, A.; Svegliati-Baroni, G., Endoplasmic Reticulum stress induces hepatic stellate cell apoptosis and contributes to fibrosis resolution. Liver Int 2012, 32, (10), 1574-84.
  8. Li, J.; Zhao, Y. R.; Tian, Z., Roles of hepatic stellate cells in acute liver failure: From the perspective of inflammation and fibrosis. World J. Hepatol. 2019, 11, (5), 412-420.

Reviewer 3 Report

All my comments were addressed. I have no further comment.

Author Response

Thank you for your letter and for your comments concerning our manuscript

We made a little revision in the writing and the references. We used “track changes” to mark up all the revisions so that you can view any changes we did to improve our manuscript quality.

We appreciate your warm work earnestly, and hope that the revised manuscript is now suitable for publication in your esteemed journal.